# Mechanism design augmented with output advice

**George Christodoulou**
Aristotle University of Thessaloniki
and Archimedes/RC Athena, Greece
gichristo@csd.auth.gr

**Alkmini Sgouritsa**
Athens University of Economics and Business
and Archimedes/RC Athena, Greece
alkmini@aueb.gr

**Ioannis Vlachos**
Athens University of Economics and Business
and Archimedes/RC Athena, Greece
ioa.vlahos@aueb.gr

## Abstract

Our work revisits the design of mechanisms via the *learning-augmented* frame-work. In this model, the algorithm is enhanced with imperfect (machine-learned) information concerning the *input*, usually referred to as prediction. The goal is to design algorithms whose performance degrades gently as a function of the prediction error and, in particular, perform well if the prediction is accurate, but also provide a worst-case guarantee under any possible error. This framework has been successfully applied recently to various mechanism design settings, where in most cases the mechanism is provided with a prediction about the *types* of the agents.

We adopt a perspective in which the mechanism is provided with an *output recommendation*. We make no assumptions about the quality of the suggested outcome, and the goal is to use the recommendation to design mechanisms with low approximation guarantees whenever the recommended outcome is reasonable, but at the same time to provide worst-case guarantees whenever the recommendation significantly deviates from the optimal one. We propose a generic, universal measure, which we call *quality of recommendation*, to evaluate mechanisms across various information settings. We demonstrate how this new metric can provide refined analysis in existing results.

This model introduces new challenges, as the mechanism receives limited information comparing to settings that use predictions about the types of the agents. We study, through this lens, several well-studied mechanism design paradigms, devising new mechanisms, but also providing refined analysis for existing ones, using as a metric the quality of recommendation. We complement our positive results, by exploring the limitations of known classes of strategyproof mechanisms that can be devised using output recommendation.

## 1 Introduction

Motivated by the occasionally overly pessimistic perspective of worst-case analysis, a recent trend has emerged focusing on the design and analysis of algorithms within the so-called *learning-augmented framework* (refer to [30] for an overview). Within this framework, algorithms are enhanced with imperfect information about the input, usually referred to as *predictions*. These predictions can stem from machine learning models, often characterized by high accuracy, leading to exceptional performance. However, their accuracy is not guaranteed, so the predicted input may differ significantly from the actual input. Blindly relying on these predictions can have significant consequences compared to employing a worst-case analysis approach.

38th Conference on Neural Information Processing Systems (NeurIPS 2024).

The framework aims to integrate the advantages of both approaches. The goal is to use these predictions to design algorithms whose performance degrades gently as a function of the inaccuracy of the prediction, known as the prediction error. In particular, they should perform well whenever the prediction is accurate –a property known as *consistency*– and also provide a worst-case guarantee under any possible error –a property known as *robustness*.

Xu and Lu [39] and Agrawal et al. [2] applied the learning-augmented framework in mechanism design settings, where there is incomplete information regarding the preferences (or types) of the participants over a set of alternatives. Traditional mechanism design addresses this information gap by devising strategyproof mechanisms that offer appropriate incentives for agents to report their true types. In the learning-augmented model, it is generally assumed that the mechanism is equipped with predictions about the types of the agents. The aim is to leverage these predicted types to design strategyproof mechanisms that provide consistency and robustness guarantees. Since then, this model has found application in diverse mechanism design settings [8, 11, 27, 25].

**Mechanisms with output advice**   In this work, we propose an alternative perspective on mechanism design with predictions. We assume that the mechanism is provided with external advice to *output a specific outcome*, rather being provided with predictions of the agents' types. For example, in a job scheduling problem, the designer may receive a recommended partition of tasks for the machines, rather than a prediction about the machines' processing times. Similarly, in an auction setting, an allocation of goods is provided, rather than a prediction about the agents' valuations.

Following the tradition of the learning-augmented framework, we make no assumptions about the quality of the recommended outcome, which may or may not be a good fit for the specific (unknown) input. The goal is to use the recommendation to design a strategyproof mechanism with good approximation guarantees whenever the recommended outcome is a good fit, but at the same time provide worst-case guarantees whenever the recommendation deviates from the optimal one.

We observe that one can reinterpret previous models within the framework of our model, viewing it as a more constrained version of predictions with *limited information*.[1] Since we only require limited information regarding the outcome, our model may be better suited to handle cases where historical input data is absent or limited, which may occur for various reasons such as privacy concerns, data protection, challenges in anonymizing, or simply because the information is missing. For instance, historical data in an auction may sometimes only contain information about the winners and perhaps the prices, omitting details about their exact valuation or the values of those who lost. Additionally, our model may be applied in cases where the designer does not need to know the specifics of the algorithm and treats it as a black box, as long as it yields satisfactory allocations, even if the inner workings are not fully understood.

We make no assumption about *how* the outcome recommendation was produced, which makes it quite general and adaptable to different application domains. For instance, the outcome may represent the optimal allocation with respect to predicted data (as seen in [2]), or a solution generated by an approximation algorithm or a heuristic. Consequently, the quality of the recommended outcome may be affected by various factors, such as the accuracy of the predicted data or limited computational resources which prevent the computation of optimal solutions, even when the data is accurate.

A beneficial side effect of our model is that an outcome recommendation fits in a plug-and-play fashion with a generic machinery for strategyproofness in multi-dimensional mechanism design, particularly maximal in range VCG mechanisms (or more generally with affine maximizers) in a straightforward manner: we simply add the recommended outcome to the range of the affine maximizer (see Section 5).

**Quality of recommendation**   In the learning augmented framework, the performance of an algorithm (or mechanism) is evaluated based on the *prediction error*, which quantifies the disparity between the predicted and actual data. Unfortunately, there is no universal definition for such an error; it is typically domain-specific (e.g., the ratio of processing times for scheduling [27, 8] or (normalized) geometric distance for facility location [2]). Therefore, if one modifies the information data model for a specific problem—for instance, by assuming that only a fraction or a signal of the predicted data is provided—it becomes necessary to redefine the prediction error.

---

[1]For example, in [2], it is assumed that the mechanism is provided with the optimal allocation with respect to the predicted types. Refer to the discussion in Section 3 for a comparison and differences with their model.

To address this issue, we propose a generic, universal measure that can be applied to analyze algorithms across various information settings and application domains. We define the *quality of recommendation* as the approximation ratio between the cost (or welfare) of the recommended outcome and the optimal cost (or welfare) both evaluated w.r.t the actual input. It is worth emphasizing that although the above definition aligns naturally with our information model, as we do not assume the designer is provided with predicted data, it can also be applied to richer information models with partial or even full predicted input.

We argue that it provides a unified metric for settings involving predictions, particularly when the objective is to design mechanisms (or more generally algorithms) with low approximation or competitive ratio. The disparity between predicted and actual data, captured by the predicted error, may not always be relevant and can lead to misleading evaluations; there are cases where this error may be significantly large, but the optimal solution remains largely unchanged. For example, consider the problem of makespan minimization in job scheduling (see also Section 3 for a detailed example in facility location). In [27, 8], the prediction error used is the maximum ratio of processing times, and it appears in the approximation guarantees. There are simple instances where this ratio is arbitrarily large, but the optimal allocation remains the same. Consequently, when the prediction error is incorporated into the analysis, it may lead to overly pessimistic guarantees for mechanisms that perform much better (see Section 3). Our metric avoids such pathological situations.

## 1.1 Contributions

We propose studying mechanisms augmented with output advice, a setup that utilizes limited information to provide improved approximation guarantees. Additionally, we introduce a unified metric that can provide more accurate evaluations, even for settings with richer information models. We explore the limitations of the class of strategyproof mechanisms that can be devised using this limited information across various mechanism design settings. Detailed results concerning the house allocation problem can be found in the full version of the paper. Table 1 summarizes our results.

**Facility Location**    In the facility location problem, there are $n$ agents each with a preferred location and the goal is to design a strategyproof mechanism that determines the optimal facility location based on an objective. In Section 3, we derive new approximation bounds for the facility location problem revisiting the Minimum Bounding Box and the Coordinatewise Median mechanisms defined in [2], as a function of the quality of recommendation. We provide tight bounds, and demonstrate that in some cases they outperform previous analysis with the use of a prediction error.

**Scheduling**    In Section 4, we study a scheduling problem with unrelated machines, where each machine has a cost for each job, which corresponds to the processing time of the job on the machine. Each job is assigned to exactly one machine, and the goal is to minimize the makespan having an output allocation as a recommendation. We devise a new strategyproof mechanism (Mechanism 1), that takes also as input a confidence parameter $\beta \in [1, n]$, reflecting the level of trust in the recommendation. We show that this mechanism is $(\beta + 1)$-consistent and $\frac{n^2}{\beta}$-robust (Theorem 3). Altogether, we obtain a $\min\{(\beta + 1)\hat{\rho}, n + \hat{\rho}, \frac{n^2}{\beta}\}$ upper bound on the approximation ratio, where $\hat{\rho}$ is the quality of the recommendation, that we show that is asymptotically tight (Theorem 4). We complement this positive result, by showing that, given only the outcome as advice, it is impossible to achieve a better consistency-robustness trade-off in the class of the weighted VCG mechanisms (Theorem 5).

**Combinatorial Auctions**    Next, we study combinatorial auctions given a recommended allocation (see Section 5). In the combinatorial auctions setting, there is a set of $m$ indivisible objects to be sold to $n$ bidders, who have private values for each possible bundle of items. We observe that our advice model fits nicely with the maximal in range VCG mechanisms or more generally with the affine maximizers, by preserving strategyproofness. These mechanisms provide the best known bounds for the approximation of the maximum social welfare for several classes of valuations [19, 17, 24]. By including the recommended outcome in the range of the affine maximizer, we immediately obtain 1-consistency, while maintaining the robustness guarantees of those mechanisms.

**House Allocation**    Finally, we switch to the house allocation problem. In this problem, we aim to assign $n$ houses to a set of $n$ agents in a way that ensures strategyproofness and maximizes the social welfare. We use the TTC mechanism [35] with the recommendation as an initial endowment, and prove that this is $\min\{\hat{\rho}, n\}$-approximate for unit-range valuations and $\min\{\hat{\rho}, n^2\}$-approximate

Table 1: Contribution Results. Consistency, robustness and approximation results proved for the mechanism design problems augmented with *output* advice. In the house allocation problem, bounds are shown for unit-range valuations, while the ones in parentheses are for unit-sum valuations. In combinatorial auctions, $\rho_M$ is the approximation ratio guarantee of a maximal in range mechanism.

| Problem | Cons | Rob | $f(\mathbf{t}, \hat{\rho})$-approximation |
|---|---|---|---|
| Facility Location (egalitarian) | 1 [2] | $1+\sqrt{2}$ [2] | $\min\{\hat{\rho}, 1 + \sqrt{2}\}$ |
| Facility Location (utilitarian) | $\frac{\sqrt{2\lambda^2+2}}{1+\lambda}$ [2] | $\frac{\sqrt{2\lambda^2+2}}{1-\lambda}$ [2] | $\min\{\sqrt{2}\hat{\rho}, \hat{\rho} + \sqrt{2}, \frac{\sqrt{2\lambda^2+2}}{1-\lambda}\}$ |
| Scheduling | $\beta + 1$ | $\frac{n^2}{\beta}$ | $\min\{(\beta + 1)\hat{\rho}, n + \hat{\rho}, \frac{n^2}{\beta}\}$ |
| Combinatorial Auctions | 1 | $\rho_M$ | $\min\{\hat{\rho}, \rho_M\}$ |
| House Allocation | 1 | $n$ (or $n^2$) | $\min\{\hat{\rho}, n \text{ (or } n^2)\}$ |

for unit-sum valuations, where $\hat{\rho}$ is the quality of recommendation. Finally, we prove it is optimal among strategyproof, neutral and nonbossy mechanisms using the characterization of [38] and the correspondence between serial dictator mechanisms and TTC mechanisms [1].

## 1.2 Related Work

**Learning-augmented mechanism design**     Recently, there has been increased interest in leveraging predictions to improve algorithms' worst case guarantees. The influential framework of Lykouris and Vassilvitskii [28] is applied on caching, formally introducing the notions of consistency and robustness, under minimal assumptions on the machine learned oracle. The learning-augmented framework is naturally brought to the algorithmic mechanism design field by [2] and [39] independently. Agrawal et al. [2] design learning-augmented strategyproof mechanisms for the problem of facility location with strategic agents. Xu and Lu [39] apply the algorithmic design with predictions framework on revenue-maximizing single-item auctions, frugal path auctions, scheduling, and two-facility location. Another version of the facility location problem, obnoxious facility location, is studied by Istrate and Bonchis [25]. Prasad et al. [31] develop a new methodology for multidimensional mechanism design that uses side information with the dual objective of generating high social welfare and high revenue. Strategyproof scheduling of unrelated machines is studied in [8], achieving the best of both worlds using the learning-augmented framework. Revenue maximization is also considered in [9] in the online setting, while Lu et al. [27] study competitive auctions with predictions. Caragiannis and Kalantzis [11] assume that the agent valuations belong to a known interval and study single-item auctions with the objective of extracting a large fraction of the highest agent valuation as revenue. Other settings enhanced with predictions include the work of Gkatzelis et al. [22], where predictions are applied to network games and the design of decentralized mechanisms in strategic settings. In [10], the scenario includes a set of candidates and a set of voters, and the objective is to choose a candidate with minimum social cost, given some prediction of the optimal candidate.

**Facility Location**     For single facility location on the line, the mechanism that places the facility on the median over all the reported points is strategyproof and optimal for the utilitarian objective, and it achieves a 2-approximation for the egalitarian social cost, which is the best approximation achievable by any deterministic and strategyproof mechanism [32]. In the two-dimensional Euclidean space, the Coordinatewise Median mechanism achieves a $\sqrt{2}$-approximation for the utilitarian objective [29], and a 2-approximation for the egalitarian objective [23]; these approximation bounds are both optimal among deterministic and strategyproof mechanisms. In [2], they consider as a prediction the position of the facility to improve the above results. Concerning the egalitarian social cost and the two-dimensional version of the problem, they achieve perfect consistency, and a robustness of $1 + \sqrt{2}$. They also prove that their mechanism provides an optimal trade-off between robustness and consistency. Regarding the utilitarian social cost in two dimensions, they propose a deterministic mechanism achieving $\frac{\sqrt{2\lambda^2+2}}{1+\lambda}$-consistency, $\frac{\sqrt{2\lambda^2+2}}{1-\lambda}$-robustness and optimal trade-off among deterministic, anonymous, and strategyproof mechanisms.

**Scheduling**     Christodoulou et al. [15] validated the conjecture of Nisan and Ronen, and proved that the best approximation ratio of deterministic strategyproof mechanisms for makespan minimization for $n$ unrelated machines is $n$. Even if we allow randomization, the best known approximation guarantee achievable by a randomized strategyproof mechanism is $O(n)$ [13]. Following the prediction framework, Xu and Lu [39] study the problem with predictions $\hat{t}_{ij}$ denoting the predicted

processing time of job $j$ by machine $i$. They propose a deterministic strategyproof mechanism with an approximation ratio of $O(\min\{\gamma\eta^2, \frac{m^3}{\gamma^2}\})$, where $\gamma \in [1, m]$ is a configurable consistency parameter and $\eta \geq 1$ is the prediction error. Balkanski et al. [8] extend these results by identifying a deterministic strategyproof mechanism that guarantees a constant consistency with a robustness of $2n$, achieving the best of both worlds.

**Combinatorial Auctions**   An important direction in combinatorial auctions related to our work is the design of strategyproof mechanisms that approximate the optimal social welfare using polynomially many queries, see e.g. [17, 24, 19, 18]. Auctions incorporating predictions have been explored across various settings such as revenue maximization auctions [11, 39], competitive auctions [27] and the online setting [9]. It is noteworthy that the design of strategyproof, near-optimal auctions using neural networks [20, 36] has been studied extensively for automated mechanism design.

**House Allocation**   Regarding the house allocation problem, Filos-Ratsikas et al. [21] prove that a randomized mechanism, called the Random Priority Mechanism, has approximation ratio of $\Theta(\sqrt{n})$, and that this is optimal among all strategyproof mechanisms. There exist lower bounds for all deterministic strategyproof mechanisms which are $\Omega(n^2)$ for unit-sum and $\Omega(n)$ for unit-range, respectively. To the best of our knowledge there is no single point of reference, for these bounds, but can follow from known results in the literature, after observing that deterministic strategyproof mechanisms are ordinal, see [14, 4]. A lower bound of $\Omega(n^2)$ on the *Price of Anarchy* for any deterministic mechanism (not necessarily strategyproof) is proved in [14]. In [4], a $\Theta(n^2)$ bound is proved for the distortion of all *ordinal* deterministic mechanisms.

## 2   Model

We consider various mechanism design scenarios that fall into the following abstract mechanism design setting. There is a set of $n$ agents and a (possibly infinite) set of alternatives $\mathcal{A}$. Each agent $i \in \{1, \ldots, n\}$ can express their preference over the set of alternatives via a valuation function $t_i$ which is private information known only to them (also called the *type* of agent $i$). The set $\mathcal{T}_i$ of possible types of agent $i$ consists of all functions $b_i : \mathcal{A} \to \mathbb{R}$. Let also $\mathcal{T} = \times_{i \in N} \mathcal{T}_i$ denote the space of type profiles.

A mechanism defines for each agent $i$ a set $\mathcal{B}_i$ of available strategies the agent can choose from. We consider *direct revelation* mechanisms, i.e., $\mathcal{B}_i = \mathcal{T}_i$ for all $i$, meaning that the agents' strategies are to simply report their types to the mechanism. Each agent $i$ provides a *bid* $b_i \in \mathcal{T}_i$, which may not match their true type $t_i$, if this serves their interests. A mechanism $(f, p)$ consists of two parts:

**A selection algorithm:**   The selection algorithm $f$ selects an alternative based on the agents' inputs (bid vector) $b = (b_1, \ldots, b_n)$. We denote by $f(\mathbf{b})$ the alternative chosen for the bid vector $\mathbf{b} = (b_1, \ldots, b_n)$.

**A payment scheme:**   The payment scheme $p = (p_1, \ldots, p_n)$ determines the payments, which also depend on the bid vector $\mathbf{b}$. The functions $p_1, \ldots, p_n$ represent the payments that the mechanism hands to each agent, i.e., $p_i : \mathcal{T} \to \mathbb{R}$.

The *utility* $u_i$ of an agent $i$ is the *actual* value they gain from the chosen alternative minus the payment they have to pay, $u_i(\mathbf{b}) = t_i(f(\mathbf{b})) - p_i(\mathbf{b})$. We consider *strategyproof* mechanisms. A mechanism is strategyproof, if for every agent, reporting their true type is a *dominant strategy*. Formally,

$$u_i(t_i, \mathbf{b}_{-i}) \geq u_i(t_i', \mathbf{b}_{-i}), \qquad \forall i \in [n], \ t_i, t_i' \in \mathcal{T}_i, \ \mathbf{b}_{-i} \in \mathcal{T}_{-i},$$

where $\mathcal{T}_{-i}$ denotes all parts of $\mathcal{T}$ except its $i$-th part.

In some of our applications (e.g. facility location and scheduling settings), it is more natural to consider that the agents are cost-minimizers rather than utility-maximizers. Therefore, for convenience we will assume that each agent $i$ aims to minimize a cost function rather than maximizing a utility function. We stress that some of our applications (e.g. facility location, one-sided matching) fall into mechanism design without money, In those cases we will assume $p_i(\mathbf{t}) = 0, \forall \mathbf{t}$ and $i \in [n]$.

**Social objective**   We assume that there is an underlying objective function that needs to be optimized. We consider both *cost minimization* social objectives (facility location in Section 3, scheduling in Section 4) and *welfare maximization* (house allocation in Section **??**, auctions in Section 5). In the context of a cost minimization problem, we assume that we are given a social cost function

$C : \mathcal{T} \times \mathcal{A} \to \mathbb{R}_+$. If all agents' types were known, then the goal would be to select the outcome $a$ that minimizes $C(\mathbf{t}, a)$.

The quality of a mechanism for a given type vector $\mathbf{t}$ is measured by the cost $\text{MECH}(\mathbf{t})$ achieved by its selection algorithm $f$, $\text{MECH}(\mathbf{t}) = C(\mathbf{t}, f(\mathbf{t}))$, which is compared to the optimal cost $\text{OPT}(\mathbf{t}) = \min_{a \in \mathcal{A}} C(\mathbf{t}, a)$. We denote an optimal alternative for a given bid vector $\mathbf{t}$ by $a^*$.

In most application domains, it is well known that only a subset of algorithms can be selection algorithms of strategyproof mechanisms. In particular, no mechanism's selection algorithm is optimal for every $t$, prompting a natural focus on the approximation ratio of the mechanism's selection algorithm. A mechanism is *$\rho$-approximate*, for some $\rho \geq 1$, if its selection algorithm is $\rho$-approximate, that is, if $\rho \geq \frac{\text{MECH}(\mathbf{t})}{\text{OPT}(\mathbf{t})}$ for all possible inputs $\mathbf{t}$.

**Mechanisms with advice** We assume that in addition to the input bid $\mathbf{b}$, the mechanism is also given as a recommendation/advice, a predicted alternative $\hat{a} \in \mathcal{A}$, but without any guarantee of its quality[2]. A natural requirement, known as *consistency*, requires that whenever the recommendation is accurate, then the mechanism should achieve low approximation. A mechanism is said to be *$\beta$-consistent* if it is $\beta$-approximate when the prediction is accurate, that is, the predicted outcome $\hat{a}$ is optimal for the given $\mathbf{t}$ vector. On the other hand, if the prediction is poor, *robustness* requires that the mechanism retains some reasonable worst-case guarantee. A mechanism is said to be *$\gamma$-robust* if it is $\gamma$-approximate for all predictions:

$$\max_{\mathbf{t}} \frac{\text{MECH}(\mathbf{t}, a^*)}{\text{OPT}(\mathbf{t})} \leq \beta \,; \qquad \max_{\mathbf{t}, \hat{a}} \frac{\text{MECH}(\mathbf{t}, \hat{a})}{\text{OPT}(\mathbf{t})} \leq \gamma \,.$$

In order to measure the quality of the prediction, we define the *recommendation error*, denoted by $\hat{\rho}$, as the approximation ratio of the recommended outcome cost to the optimal one i.e., $\hat{\rho} = \frac{C(\mathbf{t}, \hat{a})}{\text{OPT}(\mathbf{t})}$.

In some of our applications, the social objective is a welfare maximization problem, where there is an underlying welfare function $W : \mathcal{T} \times \mathcal{A} \to \mathbb{R}_+$ that needs to be maximized. We adapt our definitions for approximation and for the prediction error accordingly. In particular, the quality of a mechanism for a given type vector $\mathbf{t}$ is measured by the welfare $\text{MECH}(\mathbf{t}, \hat{a}) = W(\mathbf{t}, f(\mathbf{t}, \hat{a}))$, which is compared to the optimal welfare $\text{OPT}(\mathbf{t}) = \max_{a \in \mathcal{A}} W(\mathbf{t}, a)$. A mechanism is *$\rho$-approximate*, if $\rho \geq \frac{\text{OPT}(\mathbf{t})}{\text{MECH}(\mathbf{t})}$ for all possible inputs $\mathbf{t}$. Consistency and robustness are defined similarly to the cost minimization version, while the recommendation error is defined as the approximation ratio $\hat{\rho} = \frac{\text{OPT}(\mathbf{t})}{W(\mathbf{t}, \hat{a})}$. Note that for both versions, the quality of recommendation $\hat{\rho}$ exceeds 1, with 1 indicating perfect quality and higher values indicating poorer quality. Additionally, we require a smooth decay of the approximation ratio as a function of the quality of the recommendation as it moves from being perfect to being arbitrarily bad. We say that an algorithm is *smooth* if its approximation ratio degrades at a rate that is at most linear in $\hat{\rho}$ [5, 6, 33].

## 3 Facility Location

In this section, we study mechanisms for the facility location problem in the two-dimensional Euclidean space. There are $n$ agents each with a preferred (private) location $z_i = (x_i, y_i), 1 \leq i \leq n$ in $\mathbb{R}^2$. The goal of the mechanism is to aggregate the preferences of the agents and determine the optimal facility location at a point $f(\mathbf{t})$ in $\mathbb{R}^2$. Given a facility at point $a \in \mathbb{R}^2$, the private cost $t_i(a)$ of each agent is measured by the distance of $z_i$ from $a$, i.e., $t_i(a) = d(z_i, a)$, and the private objective of each agent is to minimize their cost. Two different social cost functions have been used to evaluate the quality of a location $a$ [2]; the *egalitarian cost*, which measures the maximum cost incurred by $a$ among all agents $C(\mathbf{t}, a) = \max_i t_i(a)$, and the *utilitarian* cost, which considers the sum of the individual costs i.e., $C(\mathbf{t}, a) = \sum_i t_i(a)$.

We assume that the mechanism is equipped with a recommended point $\hat{a} \in \mathbb{R}^2$. This is perceived as a recommendation to place the facility at $\hat{a}$. For a given $\mathbf{t}$ we denote by $a^*(\mathbf{t})$ the optimal location minimizing the social cost, and by $\hat{\rho}(\mathbf{t})$ the quality of the recommended outcome, which is defined as

---

[2]We adapt the notation accordingly to incorporate the recommendation $\hat{a}$, e.g. the selected alternative is now denoted by $f(\mathbf{t}, \hat{a})$, and the cost of the mechanism by $\text{MECH}(\mathbf{t}, \hat{a})$ etc.

the approximation ratio $C(\mathbf{t}, \hat{a})/\text{OPT}(\mathbf{t})$ and measures the approximation that would by achieved by placing the facility at $\hat{a}$. We use the simpler notation $a^*$ and $\hat{\rho}$ when $\mathbf{t}$ is clear from the context.

We note that for this problem our model coincides with the model studied in [2] for facility location problems, although our perspective is slightly different. Their paper considers that the missing information is the type of the agents, and they assume that they receive a *signal of the predicted input* $\hat{a}$, the optimal location w.r.t. the predicted types. Due to this perspective, they defined as *prediction error* the (normalized) distance of their prediction, comparing to the optimal solution w.r.t the actual types. We perceive $\hat{a}$ as an output advice. Clearly, one can interpret the output as a signal of some sort of predicted data. However, we treat the advice as a recommendation, with unknown quality, and under this perspective in the context of this paper, it makes more sense to measure it by the approximation ratio w.r.t the actual (but unknown) input.

We showcase this effect in the following example of the facility location problem in the line for the utilitarian social cost, and we further discuss it in Section 3.3. Consider $2m - 1$ agents, see Figure 1, whose preferred locations are clustered in two different points, the one at position $(0, 0)$ and the other at position $(1, 0)$, where the first point is preferred by $m$ agents and the other is preferred by $m - 1$ agents. The solution $a^*$ that minimizes the social cost places the facility at point $(0, 0)$ (preferred by $m$ agents) resulting in a total cost of $\text{OPT}(\mathbf{t}) = m - 1$. Now, take two different recommendations $\hat{a}_1$ and $\hat{a}_2$ at points $(-1, 0)$ and $(1, 0)$ respectively. The prediction error is the same for both points and it is equal to $\frac{1}{m-1}$. However, any recommendation between $a^*$ and $\hat{a}_2$ is almost optimal for large $m$, in contrast to $\hat{a}_1$. The quality of the recommendation captures this difference: the social cost for the two recommendations are $C(\hat{a}_1) = 3m - 2$ and $C(\hat{a}_2) = m$, and therefore the quality of the recommendation for $\hat{a}_1$ and $\hat{a}_2$ are respectively $\hat{\rho}_1 = \frac{3m-2}{m-1}$ and $\hat{\rho}_2 = \frac{m}{m-1}$, which converge to 3 and 1 respectively as $m$ grows.

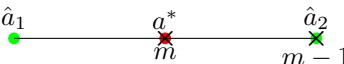

Figure 1: Quality of recommendation versus prediction error

In Section 3.1, we study the egalitarian cost and show that the Minimum Bounding Box Mechanism, defined by Agrawal et al. [2], achieves an approximation ratio of $\hat{\rho}$, which combined with the robustness bound of [2] gives an overall approximation guarantee of $\min\{\hat{\rho}, \sqrt{2} + 1\}$. In Section 3.2 we focus on the utilitarian cost and show that the Coordinatewise Median Mechanism with predictions, defined in [2], achieves an approximation ratio of at most $\sqrt{2}\hat{\rho}$ which combined with the robustness bound of [2] gives an overall approximation guarantee of $\min\{\sqrt{2}\hat{\rho}, \frac{\sqrt{2\lambda^2 + 2}}{1-\lambda}\}$, where $\lambda \in [0, 1)$ is a parameter that models the confidence of the designer on the recommendation; larger values of $\lambda$, correspond to increased confidence about the advice. Finally, in Section 3.3 we compare the bounds obtained as a function of $\hat{\rho}$ to previously known results obtained as a function of the prediction error.

## 3.1 Egalitarian Cost

The main result of this section is an approximation ratio of $\hat{\rho}$ for the egalitarian cost, by analyzing the Minimum Bounding Box mechanism defined in [2]. The robustness result for this mechanism [2], gives a total approximation ratio of $\min\{\hat{\rho}, \sqrt{2} + 1\}$, which we prove that is tight in the full version.

Intuitively, the Minimum Bounding Box mechanism works as follows[3]: If the minimum rectangle that contains all the input points $z_i, i \in \{1, \ldots, n\}$, contains the recommendation point $\hat{a}$, then we output $\hat{a}$. Otherwise, we select the boundary point with the minimum distance from $\hat{a}$.

**Theorem 1.** *The Minimum Bounding Box mechanism is* $\min\{\hat{\rho}, \sqrt{2} + 1\}$-*approximate.*

*Proof.*

$$\text{MECH}(\mathbf{t}, \hat{a}) = \max_i d(z_i, f(\mathbf{t}, \hat{a})) \leq C(\mathbf{t}, \hat{a}) = \hat{\rho}\text{OPT}(\mathbf{t})$$

The inequality holds because, whenever the prediction is outside the minimum bounding box, the mechanism projects the prediction on its boundaries, in a way that improves the egalitarian loss compared to the initial prediction. When the prediction is inside the bounding box, then $f(\mathbf{t}, \hat{a}) = \hat{a}$

and the inequality holds with equality. The term $(\sqrt{2}+1)$ follows from the robustness guarantee proved in [2]. By selecting the minimum of the two bounds, we get the approximation above. □

**Remark 1.** *We remark that when $f(\mathbf{t}, \hat{a}) = \hat{a}$, the upper bound of $\hat{\rho}$ is tight. In practice, this happens whenever the recommendation is inside the minimum bounding box defined by the agents' locations.*

### 3.2 Utilitarian Cost

Next, we show a $\sqrt{2}\hat{\rho}$ upper bound for the utilitarian cost by using the Coordinatewise Median with predictions mechanism defined in [2]. This mechanism specifies a parameter $\lambda \in [0, 1)$ which models how much the recommendation is trusted. Intuitively,[3] the mechanism works as follows; it creates $\lfloor \lambda n \rfloor$ copies of the recommendation $\hat{a} = (x_{\hat{a}}, y_{\hat{a}})$. Then, by treating each coordinate separately, it selects the median point among $n + \lfloor \lambda n \rfloor$ in total points; the $n$ actual bids $z_i = (x_i, y_i)$ and the $\lfloor \lambda n \rfloor$ copies of the recommendation. After calculating the medians $x_a$ and $y_a$ for each coordinate, it defines the outcome to be $f(\mathbf{t}, \hat{a}) = (x_a, y_a)$. In the full version of the paper, we show that our analysis is tight.

**Theorem 2.** *The Coordinatewise Median with Predictions mechanism is $\min\{\sqrt{2}\hat{\rho}, \hat{\rho}+\sqrt{2}, \frac{\sqrt{2\lambda^2+2}}{1-\lambda}\}$- approximate.*

### 3.3 Comparison of Error Functions

In this section, we compare the quality of recommendation $\hat{\rho}$ to the error $\eta$ defined in [2] and find instances for which our bounds are tight while previous known bounds are not. We first establish that $\hat{\rho} \leq \eta + 1$ holds for both the egalitarian and the utilitarian objective. We then show that for both objectives, there exist instances that our bounds are strictly better than the ones proved in [2].

**Lemma 1.** *For the egalitarian social cost, there exists an instance where $\hat{\rho} < \eta + 1$.*

**Lemma 2.** *For the utilitarian social cost, there exists an instance where $\sqrt{2}\hat{\rho} < \frac{\sqrt{2\lambda^2+2}}{1+\lambda} + \eta$*

We give all the proofs in the full version of the paper. Note that for the egalitarian objective, our bound is a refinement of the (tight) bound $\eta + 1$ from [2]. On the other hand, for the utilitarian objective, there exist instances for which the $\frac{\sqrt{2\lambda^2+2}}{1+\lambda} + \eta$ bound of [2] is better than ours. For this reason, in the full version of the paper we observe the behaviour of $\hat{\rho}, \eta$ in real-world datasets [26, 7, 37, 12, 16, 3].

## 4 Scheduling

In this section, we study strategyproof mechanisms for the *makespan minimization scheduling problem*. In this problem, we have a set $N$ of $n$ unrelated machines (the agents) and a set $M$ of $m$ jobs. Each machine $i$ has a (private) cost $t_{ij}$ for each job $j$, which corresponds to the processing time of job $j$ in machine $i$. Since we consider only strategyproof mechanisms, each machine $i$ declares their *true* cost $t_{ij}$ for each job $j$; let $t_i = (t_{i1}, \ldots, t_{im})$. The goal of the mechanism is to process the machines' declarations $\mathbf{t} = (t_1, \ldots, t_n)$ and subsequently determine both an allocation $a(\mathbf{t})$ of the jobs to the machines and a payment scheme $p(\mathbf{t}) = (p_1(\mathbf{t}) \ldots, p_n(\mathbf{t}))$, where $p_i(\mathbf{t})$ is given to each machine $i$ for processing their allocated jobs. An allocation is given by a vector $a = (a_1, \ldots, a_n)$, where $a_i = (a_{i1}, \ldots, a_{im})$, and $a_{ij}$ is set to 1 if job $j$ is assigned to machine $i$ and 0 otherwise. An allocation $a$ is feasible if each job is allocated to exactly one machine, i.e., $\sum_{i \in N} a_{ij} = 1$, for all $j \in M$, and $\sum_{i \in N, j \in M} a_{ij} = m$; we denote by $\mathcal{A}$ the set of all feasible allocations.

The cost experienced by each machine $i$ under an allocation $a$ is the total cost of all jobs assigned to it: $t_i(a) = t_i(a_i) = \sum_{j \in M} t_{ij} a_{ij} = t_i \cdot a_i$. The private objective of each machine $i$ is to maximize their utility $u_i(\mathbf{t}) = p_i(\mathbf{t}) - t_i(a(\mathbf{t}))$. In the strategyproof mechanisms that we consider here, this happens when each machine declares its true cost. The social cost function that is usually used in this problem in order to evaluate the quality of an allocation $a$, is the maximum cost among all machines, which is known as the makespan: $C(\mathbf{t}, a) = \max_i t_i(a)$.

---

[3]For completeness we include the definition of the mechanism in the full version. We further refer the reader to [2] for the exact definition and for the proof of strategyproofness and robustness.

We assume that the mechanism is provided with a recommendation $\hat{a} \in \mathcal{A}$, which can be seen as a suggestion on how to allocate the jobs to the machines. For a given $\mathbf{t}$ we denote by $a^*(\mathbf{t})$ the optimal allocation minimizing the social cost function, i.e., $a^*(\mathbf{t}) \in \arg\min_{a \in \mathcal{A}} C(\mathbf{t}, a)$, and by $\text{OPT}(\mathbf{t})$ the minimum social cost, i.e., $\text{OPT}(\mathbf{t}) = C(\mathbf{t}, a^*(\mathbf{t}))$. We measure the quality of the recommended outcome with $\hat{\rho}(\mathbf{t})$, which is defined as the approximation ratio $C(\mathbf{t}, \hat{a})/\text{OPT}(\mathbf{t})$ and measures the approximation that we would achieve if we selected the recommended allocation $\hat{a}$. In the notation of $a^*$ and $\hat{\rho}$, we drop the dependency on $\mathbf{t}$ when it is clear from the context.

In the remainder of this section, we introduce a strategyproof mechanism that we call Allocation-ScaledGreedy (Mechanism 1). We prove that, given a confidence parameter $1 \le \beta \le n$, it exhibits $(\beta+1)$-consistency and $\frac{n^2}{\beta}$-robustness (Theorem 3). Next, we investigate the smoothness of this mechanism and demonstrate that its approximation ratio is upper bounded by $\min\{(\beta+1)\hat{\rho}, n+\hat{\rho}, \frac{n^2}{\beta}\}$, which is asymptotically tight (Theorem 4). Furthermore, we establish that, when provided with the outcome as advice, it is impossible to achieve a better consistency-robustness trade-off than the AllocationScaledGreedy mechanism within the class of weighted VCG mechanisms (Theorem 5).

## 4.1 AllocationScaledGreedy Mechanism

In this subsection, we introduce a strategyproof mechanism called AllocationScaledGreedy, which achieves a $(\beta+1)$-consistency (more precisely, $(\frac{n-1}{n}\beta+1)$-consistency which converges to $\beta+1$ for large $n$) and a $\frac{n^2}{\beta}$-robustness, where $\beta$ is a confidence parameter ranging from 1 to $n$, with 1 corresponding to full trust and $n$ corresponding to mistrust. For $\beta = n$, which can be interpreted as ignoring the recommendation, the AllocationScaledGreedy mechanism corresponds to the VCG mechanism; in that case, consistency and robustness bounds coincide, giving an $n$-approximation (same as VCG). Regarding the smoothness of our mechanism, we prove an asymptotically tight approximation ratio of $\min\{(\beta+1)\hat{\rho}, n+\hat{\rho}, \frac{n^2}{\beta}\}$.

**AllocationScaledGreedy**  The mechanism sets a weight $r_{ij}$ for every machine $i$ and every job $j$ based on the recommendation $\hat{a}$. $r_{ij}$ is set to 1 wherever $\hat{a}_{ij} = 1$, and $\frac{n}{\beta}$ wherever $\hat{a}_{ij} = 0$, for some $\beta \in [1, n]$. It then decides the allocation by running the weighted VCG mechanism for each job $j$ separately, and by using $r_{ij}$ as the (multiplicative) weight of machine $i$, i.e., each job $j$ is allocated to some machine in $\arg\min_i\{r_{ij}t_{ij}\}$ that we denote by $i_j$.

---

**Mechanism 1** The AllocationScaledGreedy mechanism

---

**Input:** instance $\mathbf{t} \in \mathbb{R}^{n \times m}$, recommendation $\hat{a} \in \mathbb{R}^{n \times m}$
**Output:** $a$
  1: $r_{ij} \leftarrow 1$ if $\hat{a}_{ij} = 1$, $\frac{n}{\beta}$ otherwise, $(\beta \in [1, n])$
  2: $i_j \leftarrow \arg\min_i\{r_{ij}t_{ij}\}$
  3: if $i = i_j$ then $a_{ij} = 1$ else $a_{ij} = 0$, for each $(i, j) \in N \times M$

---

**Remark 2.** *We remark that the AllocationScaledGreedy mechanism for $\beta = 1$ is a simplification of the SimpleScaledGreedy mechanism of [8]. In [8], it is assumed that the mechanism is equipped with predictions of the entire cost matrix $\hat{t}_{ij}$, for every machine-job pair. The SimpleScaledGreedy mechanism utilizes this information to define weights $r_{ij}$ that may take values in the range $[1, n]$. In contrast, AllocationScaledGreedy uses weights with values only 1 or $n$, for $\beta = 1$. Notably, despite the limited information available to AllocationScaledGreedy, both mechanisms share the same consistency and robustness, but SimpleScaledGreedy lacks the nice property of being smooth, as for a very small prediction error, the approximation ratio has a large discontinuity gap (see full version for an example) as opposed to AllocationScaledGreedy (Theorem 4). SimpleScaledGreedy served as an intermediate step in [8] in the design of the more sophisticated mechanism ScaledGreedy, (which again relies heavily on the prediction of the entire cost matrix) which achieves the best of both worlds, constant consistency and $O(n)$-robustness. However, for similar reasons, ScaledGreedy is not smooth either.*

**Theorem 3.** *The AllocationScaledGreedy mechanism is $\left(\frac{n-1}{n}\beta+1\right)$-consistent and $\frac{n^2}{\beta}$-robust.*

In the following theorem, we show the smoothness result for the AllocationScaledGreedy mechanism; we show a tight approximation ratio depending on $\hat{\rho}$. We prove this theorem in the lemmas. In the

first one, we show that $\min\{(\beta + 1)\hat{\rho}, n + \hat{\rho}, \frac{n^2}{\beta}\}$ is an upper bound, and in the second one that $\min\{\frac{n-1}{n}\beta\hat{\rho}, \frac{n+\hat{\rho}-1}{2}, \frac{n^2-1}{2\beta}\}$ is a lower bound on the approximation ratio of the AllocationScaled-Greedy mechanism. We defer the reader to the full version for the complete proof.

**Theorem 4.** *The AllocationScaledGreedy mechanism is at most $\min\{(\beta + 1)\hat{\rho}, n + \hat{\rho}, \frac{n^2}{\beta}\}$-approximate and this bound is asymptotically tight.*

### 4.2 Mechanism Optimality

In this subsection, we provide general impossibility results for the class of weighted VCG mechanisms[4], the most general known class of strategyproof mechanisms for multi-dimensional mechanism design settings, such as the scheduling problem. We prove that it is impossible to improve upon the AllocationScaledGreedy mechanism, given the recommended outcome. More specifically, there is no weighted VCG mechanism with $\beta$-consistency that can achieve a robustness better than $\Theta(\frac{n^2}{\beta})$, highlighting the optimality of AllocationScaledGreedy in this class of mechanisms.

**Theorem 5.** *Given any recommendation $\hat{a}$, any weighted VCG mechanism that is $\beta$-consistent, must also be $\Omega(\frac{n^2}{\beta})$-robust, for any $2 \leq \beta \leq n$.*

*Proof sketch.* We provide a proof sketch of Theorem 5 and refer the reader to the full version for the complete proof. We will consider instances with $n$ machines and $n^2$ jobs. Let a $\beta$-consistent weighted VCG mechanism and a recommendation $\hat{a}$ that assigns every $n$ jobs to a distinct machine. Focusing on each machine $i$, we specify the cost vector $\mathbf{t}$, such that the optimal allocation matches $\hat{a}$. The costs are such that the mechanism must assign each job $j$ either to machine $i$ or to machine $\hat{i}_j$ that receives job $j$ in $\hat{a}$. Machine $i$ should not receive many jobs, otherwise $\beta$-consistency is violated. Consequently, there are many (approximately $\frac{n^2}{2}$) weights $r_{ij}$ with value much higher comparing to the weight $r_{\hat{i}_j j}$, i.e., $\frac{r_{ij}}{r_{\hat{i}_j j}} \geq \frac{n}{2\beta}$.

Since this is true for each machine $i$, there exists a machine $\hat{i}$, such that, focusing only on the $n$ jobs that $\hat{i}$ receives in $\hat{a}$, there exist approximately $\frac{n^2}{2}$ jobs with value much higher (comparing to $\hat{i}$) among all machines. Then it holds that we can assign approximately $\frac{n}{2}$ jobs to distinct machines such that those machines have high-valued weight for their assigned job; let $J$ be the set of those jobs. We finally consider the instance where each of those machines has a cost of 1 for their assigned job and sufficiently high cost[5] for any other job in $J$, machine $\hat{i}$ has a cost slightly less than $\frac{n}{2\beta}$ for jobs in $J$, and all other machines have infinite cost for jobs in $J$. The cost for any other job that does not belong to $J$ is 0 for any machine. In this instance $\mathbf{t}$, $\text{OPT}(\mathbf{t}) = 1$, but the mechanism allocates all jobs of $J$ to machine $\hat{i}$, resulting in $\text{MECH}(\mathbf{t}, \hat{a})$ being approximately $\frac{n^2}{4\beta}$. Hence, any $\beta$-consistent weighted VCG mechanism is $\Omega(\frac{n^2}{\beta})$-robust.

## 5 Combinatorial Auctions

In this section, we show how output advice can integrate with truthful maximal in range (MIR) mechanisms where the goal is to optimize the social welfare (or more generally an affine function) over a restricted outcome space. Let $M$ be a MIR mechanism with an approximation guarantee $\rho_M$. We define a mechanism that compares the outcome of $M$ with a suggested solution $\hat{a}$ and selects the one that achieves the highest social welfare. This mechanism remains MIR, as it simply expands the range of possible outcomes to include $\hat{a}$, ensuring it remains strategyproof, and is $\min\{\hat{\rho}, \rho_M\}$-approximate. Combining with the results of [34, 17] we obtain strategyproof mechanisms for combinatorial auctions with approximation ratio of $\min\{\hat{\rho}, m/\log m\}$ for general valuations, $\min\{\hat{\rho}, \sqrt{m/\log m}\}$ for subadditive valuations, and $\min\{\hat{\rho}, 2\}$ for multi-unit valuations.

---

[4]Technically, weighted VCG mechanisms choose weights $r_i$ for each machine $i$, rather than the more general case of choosing $r_{ij}$ for each machine $i$ and job $j$, that we consider here. In scheduling, where the valuation domain is additive, jobs can be grouped into clusters, and a distinct VCG mechanism can be applied to each cluster. The composition of these mechanisms remains strategyproof for additive domains. The extreme (and more general) case considered here is to cluster the jobs into $m$ clusters.

[5]We choose $\infty$ cost for clarity, in fact it suffices to choose instead $t_{ij} > \frac{\min_{i'}\{r_{i'j}t_{i'j}\}}{r_{ij}}$, such that the mechanism does not allocate job $j$ to machine $i$.

## Acknowledgments and Disclosure of Funding

This work has been partially supported by project MIS 5154714 of the National Recovery and Resilience Plan Greece 2.0 funded by the European Union under the NextGenerationEU Program.

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
