# OpenReview forum: "Mechanism design augmented with output advice"
_NeurIPS.cc/2024/Conference — NeurIPS 2024 spotlight_

### Official Review · Reviewer_PWat · 2024-07-12

**Soundness:** 3
**Presentation:** 2
**Contribution:** 3
**Rating:** 6
**Confidence:** 2

**Summary:**

This paper explores a novel setting in mechanism design where an output is provided as advice to the mechanism. The authors propose consistency, robustness, and approximation properties for strategy-proof mechanisms. They introduce four types of mechanism design problems and corresponding mechanisms, demonstrating their beneficial properties.

**Strengths:**

The setting and algorithms used in mechanism design are intriguing. The approximation analysis is well-conducted and represents a significant technical contribution.

**Weaknesses:**

1. The paper is difficult to follow due to its presentation. For example, the model of this work differs from other works mentioned in the paper [2, 42]. However, these other works are discussed before the authors’ own work in the introduction, which seems redundant. Additionally, the paper often repeats similar sentences.
2. Many contributions are relegated to the appendix. This is not ideal as the main body should be self-contained, with the appendix used for further verifications.

**Questions:**

1. It is not clear why the output advice might be helpful for mechanism design and why the model requires potentially inaccurate output advice. Why can't the model compute an output from the input (type profile) directly and regard this output as advice?
2. Following the first question, do you have comparative results showing that without output advice, the consistency of the optimal strategy-proof mechanism is strictly worse than that of the strategy-proof mechanism with accurate output advice?
3. In my understanding, the VCG mechanism provides an output that maximizes social welfare (minimizes cost), so the approximation ratio of VCG should be 1 if the output is computed precisely. How do you explain your results indicating that the approximation ratio of VCG is $\Theta(n)$, which seems counterintuitive?

**Limitations:**

Although the poor presentation does not diminish the positive contributions of this paper, it is a drawback that makes the content difficult to understand.

---

> ### Author Rebuttal · Authors · 2024-08-06
>
> We thank the reviewer for their comments. We are sorry that this reviewer (unlike the other two) was not satisfied with the level of presentation of our results. We argue below why we find the reasons for the low presentation score to be a bit too harsh.
>
> **Reviewer PWat stated as two weaknesses:**
>
> *1. "The paper is difficult to follow due to its presentation. For example, the model of this work differs from other works mentioned in the paper [2, 42]. However, these other works are discussed before the authors’ own work in the introduction, which seems redundant. Additionally, the paper often repeats similar sentences."*
>
> *2. "Many contributions are relegated to the appendix. This is not ideal as the main body should be self-contained, with the appendix used for further verifications."*
>
>
> **Response to weakness 1**: In the introduction, we propose a model as an alternative of those discussed in [2,42] so we feel that is important to discuss their model *before* we introduce ours and argue why our model is motivated given those previous models.
>
> **Response to weakness 2**: We manage to include all the main contributions in the main body, however space limitations force us to state the detailed proofs in the appendix (we have prepared a full version of the paper). This is typical for a theoretical work as ours.
>
>
> **Response to questions**
>
> **Question 1**: *"It is not clear why the output advice might be helpful for mechanism design and why the model requires potentially inaccurate output advice. Why can't the model compute an output from the input (type profile) directly and regard this output as advice?"*
>
> **Our response**: We are not sure that we understand this question. In a mechanism design setting, agents' types/preferences are private (see lines 176, 236, 311), and are not offered directly to the algorithm designer. Therefore, the input (type profile) is not known. The whole goal of mechanism design is to design strategyproof mechanisms, i.e., mechanisms that provide incentives to the agents to reveal their true types (see Introduction, lines 39-48, and Model Section, lines 179-192). Therefore, we are not sure what is meant by "computing the output from the input (type profile)", as this is not offered to the designer.
>
> **Question 2**: *"Following the first question, do you have comparative results showing that without output advice, the consistency of the optimal strategy-proof mechanism is strictly worse than that of the strategy-proof mechanism with accurate output advice?"*
>
> **Our response**: Again, we are not sure that we understand this question. The definition of consistency requires some sort of advice/prediction (see lines 37, 214-220). We thoroughly compare the bounds of our mechanisms that are enhanced with output advice with the optimal mechanisms without any advice/prediction.
>
> Those comparisons for the four problems are more specifically the following:
>
> - Facility location problem: The bounds of the optimal strategyproof mechanisms without any prediction/advice is 2 for the egalitarian cost objective (lines 511-514). The consistency (i.e., its approximation ratio when the output advice is accurate) of the Minimum Bounding Box Mechanism (lines 284-286) is 1 (by setting the quality of recommendation, $\hat{ρ}$, equal to 1 in Theorem 1).
>
> - Scheduling games: The worst-case guarantee of the VCG mechanism (which is the optimal strategyproof mechanism) is the poor approximation of $n$ (lines 522-524). However, the consistency (accurate output advice) of the AllocationScaledGreedy mechanism for the scheduling problem, is asymptotically constant (Theorem 3 for $\beta$ being a constant) which is strictly better than $n$.
>
> - House allocation problem: Regarding deterministic stategyproof mechanisms, a $\Omega(n^2)$ bound is known for the unit-sum case and a $\Omega(n)$ bound is known for the unit-range case (lines 539-543). The TTC with recommended endowment mechanism (Mechanism 5, lines 832-833) is 1-consistent in both cases by setting $\hat{ρ}=1$ in Τheorem 7.
>
> - Combinatorial auctions: the worst-case approximation ratios of the best known strategyproof mechanisms for the three applications are strictly more than 1 (lines 1000, 1006, 1011), while the MIR with recommended allocation (Mechanism 6, lines 978-979) is 1-consistent (by setting $\hat{ρ}=1$ in Lemma 16), which is strictly better.
>
>
> **Question 3**: *"In my understanding, the VCG mechanism provides an output that maximizes social welfare (minimizes cost), so the approximation ratio of VCG should be 1 if the output is computed precisely. How do you explain your results indicating that the approximation ratio of VCG is $\Theta(n)$, which seems counterintuitive?"*
>
> **Our response**: It is true that the VCG mechanism by definition finds the optimal solution in the case of the social welfare (or cost) objective. However, in the scheduling game we do not study the welfare objective, but the makespan i.e., the minimization of the maximum completion time (see intro (line 98, 119), and as we define in the description of the problem (line 325)). This is the standard objective (see e.g., [9,14,16,42]) in the scheduling literature (defined in the description of the problem, line 325). Regarding the makespan objective, VCG is known to be [16] the optimal strategyproof mechanism but with a very poor approximation ratio of $n$ (as stated in lines 522-524).

---

> > ### Comment · Reviewer_PWat · 2024-08-09
> >
> > Thank you for your detailed response.
> >
> > * Regarding weakness 1
> >
> > The paper introduces the mechanism design problem with predicted inputs in the first three paragraphs.
> > In my understanding, the model does not enhance or extend the existing model but rather considers another model, that focuses on the advice in the output dimension.
> > I think that it's more appropriate to move them into *Related Work*.
> >
> > When I reviewed this paper, I felt that the model extended the existing model of mechanism design with predicted inputs until I got to *Line 47*.
> >
> > * Regarding weakness 2
> >
> > It's fine to remove the proofs from the appendix.
> > However, this paper removes the main results (regarding House Allocation and Auctions) from the appendix, which I have never seen in other conference papers.
> >
> > However, I acknowledge that the presentation score is harsh for these issues, and I improve the presentation score from 1 to 2.
> >
> > * Regarding my original question 1,2
> >
> > I originally meant that the model needs justification about whether the output advice would be helpful for designing a strategy-proof mechanism.
> >
> > The original question 1 should be stated as follows: Given a mechanism $M$ with output advice, now we construct a mechanism $M_1$ without output advice. This mechanism takes type profile $\boldsymbol{t}$ as input, compute a (possibly accurate or inaccurate) outcome advice $a$ by some oracle $O$, take $\boldsymbol{t}$ and $a = O(\boldsymbol{t})$ as input to run the mechanism $M$, and return the outcome $M(\boldsymbol{t}, a)$ to players.
> > Why can we not do such a reduction from a mechanism with output advice to a mechanism without output advice?
> >
> > For original question 2, I meant to say "the approximation ratio of the optimal strategy-proof mechanism".
> >
> > Your response to question 2 is satisfactory. I will consider improving my rating after the full responses.
> > But I suggest that these results be listed in Table 1, as a comparison of mechanisms with/without output advice and a justification of the output-advised model.
> > Besides, the comparisons of mechanisms with output advice and input predictions are also encouraged, as mentioned by *Reviewer i6E2*.
> >
> > * Minor issues
> >
> > When I look through the paper again, I found that it's better to replace $a^*$ with $a^*(t)$ in the expression between *Line 219* and *Line 220*, since $a^*$ depends on $t$ and $t$ is not fixed in the place $a^*$ appears.

---

> ### Author Response · Authors · 2024-08-13
>
> Thank you for the clarifications on your questions and for considering to increase the scoring.
>
> Regarding weakness 1, our model indeed differs from the literature and is not just an extension. Still we believe that it is important to introduce previous work on mechanism design with advice (expressed as input or output prediction) so that we can compare our model with existing literature and show how it differs from it.
>
> Regarding weakness 2, we understand the reviewer's concern, but we believe that the way we present the paper serve the following purposes: Our model's error function is justified by the facility location problem, while the model's advice type (output advice) is supported by the scheduling problem. The house allocation and combinatorial auctions sections can be viewed as applications of our model, which is why we mention the results in the contributions section but include the detailed results in the appendix.
>
> In any case, it seems that we have different (subjective) view on some presentation aspects and we thank the reviewer for their intention to increase this score.
>
> **Our response to the original question 1:**
> The challenge in the reduction proposed by the reviewer is how to define the oracle $O(t)$. If you take any arbitrary oracle, e.g. one that produces the optimal allocation (or a good approximation), then the players may have incentive to misreport, as we mention in 181-182. If one designs an oracle $O(t)$ as part of a strategyproof mechanism, this is the standard mechanism design problem *without advice or prediction*. It is known that strategyproofness imposes limitations so we cannot implement arbitrary good approximation algorithms, as we mention in the related section (see e.g scheduling [16], lines 522-524). For example, in scheduling games one cannot expect to design a strategyproof mechanism by getting an oracle $O(t)$ with good  approximation and use it in the reduction proposed by the reviewer, since this would produce a good approximation for this mechanism; this would contradict the result of [16] which states that no strategyproof mechanism has approximation ratio better than $n$. This is why in all learning-augmented mechanisms (like [2], [9] and our work) it is assumed that the prediction is an exogeneous (untrusted) source unrelated to the actual input $t$, as we mention in the text (see e.g. lines 53-55, 68-69).

---

> > ### Comment · Reviewer_PWat · 2024-08-13
> >
> > Thank you for the authors' further clarification. I am happy to improve the contribution score to 3 and the overall score to 6 (weak accept), considering that the model is indeed innovative and non-trivial.
> >
> > Regarding the presentation issue, I agree with the authors that "it is important to introduce previous work". But what I mentioned is that the presentation in this version is somewhat misleading, especially in lines 28-29: "Within this framework, algorithms are enhanced with *imperfect information about the input*, usually referred to as predictions." It (as well as other sentences) makes me feel that the paper studies the mechanism design with predicted inputs.
> > Overall, there is no conflict of view between the authors and me. Though important to introduce, I insist that the presentation logic here is not appropriate. I suggest replacing the above-mentioned sentence with, "Within this framework, one approach is to enhance the algorithm with imperfect information about the input, ... However, the algorithm enhanced with output advice lacks study."
> >
> > I also agree with the authors that there is heterogeneity in presentation preference. I do not consider the presentation conflict when I re-evaluate the overall score of this paper.

---

### Official Review · Reviewer_i3AA · 2024-07-13

**Soundness:** 3
**Presentation:** 4
**Contribution:** 4
**Rating:** 7
**Confidence:** 2

**Summary:**

The authors propose a novel paradigm for mechanism design augmented with advice. While classically learning-augmented mechanism design assumes input advice, the authors consider output/outcome advice. They use "quality of recommendation" to quantify the quality of the advice and provide approximation, consistency, and robustness guarantees as a function of confidence in the advice and quality of recommendation.

**Strengths:**

The authors propose a novel learning-augmented framework and use it to analyze various well-studied mechanism design settings, contextualizing well in the subfield of learning-augmented mechanism design.

**Weaknesses:**

I see no significant weaknesses, granted that this is not my area of expertise. Formatting in some parts of the paper can be improved (e.g., spacing and commas in line 110 and consistency of the formatting of the citations). As noted below, I was wondering if there is a reason you only conduct experiments for one of the mechanism design settings.

**Questions:**

1. Why do you only conduct experiments in one of the mechanism design settings?

**Limitations:**

Yes

---

> ### Author Rebuttal · Authors · 2024-08-06
>
> We thank the reviewer for their thorough review. We address their concern below.
>
> **Question**: *"Why do you only conduct experiments in one of the mechanism design settings?"*
>
> **Our response**: Our work is mainly theoretical, and we provide tight results for all four problems and comprehensive comparisons with the literature whenever related literature exists (facility location [2,42] and scheduling [9,42]). For the facility location problem with egalitarian cost, due to space limitation, those comparisons appear in the appendix (section B.4, see Lemmas 4 and 5, and lines 639-643). For scheduling games, the comparison is given in Theorem 5, and Remark 2. There is only one exception to that, where our theoretical comparison with the literature [2], is inconclusive, and this is regarding the facility location problem with utilitarian cost. This is the reason why we conduct experiments only for this case (Section B.5).

---

> > ### Comment · Reviewer_i3AA · 2024-08-13
> >
> > Thank you for the response. Having read the rebuttal (and the discussion with the other reviewers), I will maintain my score.

---

### Official Review · Reviewer_i6E2 · 2024-07-14

**Soundness:** 3
**Presentation:** 3
**Contribution:** 3
**Rating:** 7
**Confidence:** 3

**Summary:**

This paper studies the problem of mechanism design with prediction and introduces a new framework based on output predictions. Unlike most previous work, which primarily uses input predictions with varying error metrics, this paper considers output predictions and proposes a new error metric that can be applied in various settings. The authors reexamine four previously studied problems with this new prediction and develop consistency, robustness, and smoothness results for each setting.

**Strengths:**

1. The concept of output prediction is novel, and the error metric is general, applicable to a range of problems.
2. The results for the four settings are comprehensive, with some being tight. The paper also provides comparison to input prediction results in some settings.

**Weaknesses:**

Except for the facility location section, this paper lacks extensive comparisons to previous results with input predictions. While there are tight results with respect to the output prediction, it is not clear to me what does this mean compared to those with input predictions, and what are the relations between them. More comparison results (both theoretical and empirical) between different forms of predictions and error metrics should be conducted.

**Questions:**

refer to the weaknesses.

**Limitations:**

1. Refer to the weaknesses
2. There are some minors, mainly on the direction of inequalities. While I am not sure if my interpretation is correct, I suggest the authors double-check on these points:
    1. line 203: $(W(t,a)$ -> $C(t,a)$
    2. line 211: $\geq$ -> $\leq$
    3. equations between line 219 and line 220: $\leq$ -> $\geq$

---

> ### Author Rebuttal · Authors · 2024-08-06
>
> We thank the reviewer for their insightful comments. We address concerns about potential weaknesses below.
>
> **Question**: *"Except for the facility location section, this paper lacks extensive comparisons to previous results with input predictions. While there are tight results with respect to the output prediction, it is not clear to me what does this mean compared to those with input predictions, and what are the relations between them. More comparison results (both theoretical and empirical) between different forms of predictions and error metrics should be conducted."*
>
> **Our response**: We would like to argue that we thoroughly compare our results with related literature of this relatively new research area. In particular, we consider four different mechanism design settings. Only two of those were studied before within the learning augmented framework (facility location and scheduling) and we thoroughly compare our results with existing literature in both settings. The other two (welfare maximization in auctions and house allocation) were not previously studied using the learning augmented framework, so we would not be able to compare our work with previous results.
>
> As a matter of fact, all previous work regarding the facility location problem [2, 42] are based on output (not input) predictions, as our work does, but [42] studies multiple facilities (as we describe in the related work section), so it is not closely related, hence incomparable with our work. [2] studies the same problem and this is why we provide a thorough comparison with this work. In particular, this is the perfect setting to demonstrate why our proposed error (the quality of recommendation) is more natural and results in a better refinement of the approximation bounds (in the case of egalitarian cost).
>
> Regarding scheduling games, again we compare our work with all existing previous results [9, 42]. We make an explicit comparison only with [9] which is the state-of-the art (as [9] improves the bounds of [42] as we discuss in the further related work section that appears in Appendix A). Regarding [9], in Remark 2, we provide a thorough explanation of the connection of our AllocationScaledGreedy mechanism that is enhanced with output advice with the two mechanisms proposed in [9] that use input advice instead. Moreover, our optimality bounds (Theorem 5) provide the first trade-off between the amount of provided information and the best achievable bounds. We will follow the excellent suggestion of the reviewer to highlight this separation in the paper and make it more prominent.
>
> Regarding empirical comparison, we would like to emphasize that our work is mainly theoretical. In the facility location problem with egalitarian cost and in the scheduling problem, our theoretical results are sufficient to provide comprehensive comparison of our results with the work of [2] and [9], respectively. For the facility location problem with egalitarian cost, due to space limitation, those comparisons appear in the appendix (section B.4, see Lemmas 4 and 5, and lines 639-643). For scheduling games, the comparison is given in Theorem 5, and Remark 2. The only case where our theoretical comparison with the results of [2] is inconclusive is the facility location problem with utilitarian cost, and this is the reason why we conduct experiments only for this case (Section B.5).
>
> We thank the reviewer for catching some typos. The correction on line 203 is indeed correct. The inequalities however should stand as they are, as they refer to the cost minimization objective.

---

> > ### Comment · Reviewer_i6E2 · 2024-08-10
> >
> > Thanks for the response. the writing style of the introduction makes me expect a detailed comparison with the previous results with input prediction. Now it seems that only the scheduling games have been studied in the learning augmented literature with input advice. I guess my original questions are a bit general and beyond the scope of this paper:
> > 1. Given algorithms with input predictions and output predictions, respectively,  each having an approximation ratio depending on the respective prediction error defined, how can we compare these two algorithms and their performance?
> >
> >
> > Here are several other further questions with respect to the contents of the paper:
> > 1. Given an input prediction, we can compute the corresponding output and apply the mechanism proposed in your paper. As the input prediction contains more information beyond the output, can we possibly design better algorithms beyond that?
> > 2. Several tight results are with respect to the proposed mechanisms (e.g., Lemma 1, Lemma 4, Lemma 9). Can these tight results imply tightness results to the problem itself under the arbitrary mechanisms?
> > 3. With respect to Lemma 7, can there be instances where $\sqrt{2} \hat{\rho} > \frac{\sqrt{2 \lambda^2+2}}{1+\lambda}+\eta$?
> > 4. As for the reason that mechanism design with output advice in MIR mechanisms has such plug-and-play property, is it because that computing an output from an input prediction can be computationally hard? It is quite like the trivial idea that you can get a mechanism with good expected consistency and robustness by just randomizing over a robust mechanism and a mechanism that fully trusts the prediction.
> > 5. The proofs of tightness results mainly propose instances that make the ratio tight. I am wondering, even if such an instance exists, can we still find ratio that equals the tight value among these instances, and have more tight ratios among other instances? If this holds, can we still say those ratios are tight?
> >
> > Further minor comments
> > 1. A new typo I just find: line 206: $max_{a\in\mathcal{A}}C(t,a)$ -> $min_{a\in\mathcal{A}}C(t,a)$
> > 2. several prediction errors used in the previous work and definitions are lacking (e.g., the MIR mechanism), which cause trouble for further reading.

---

> ### Author Response · Authors · 2024-08-13
>
> Thank you for your comments and follow-up questions. We respond to these questions below:
>
> **Question**: *Given algorithms with input predictions and output predictions, respectively, each having an approximation ratio depending on the respective prediction error defined, how can we compare these two algorithms and their performance?*
>
> **Our response**: Indeed, this is an excellent, deep and quite general question, that could be explored in future work. Even comparing algorithms with *different* input predictions can be challenging and interesting, and comparison may vary according to the respected prediction error. We emphasize that it is not our intention (and beyond the scope of our work) to do such a general comparison so we only do, as this reviewer noted, whenever there is existed related work, i.e. for the facility location problem and for the scheduling problem.
>
> ## Further questions:
>
> **Our response to Question 1:** Indeed, the reviewer is right. As we mention in lines 58-59 and 69-70, one can see our model as a restriction of models that have input predictions, which makes the design of mechanisms much more challenging comparing to those models that have input predictions. The scheduling problem is an excellent example for this discrepancy. The ScaledGreedy mechanism [9] that uses the input as a prediction can achieve a constant consistency together with a linear robustness (see Remark 2, lines 363-365, and Related Work, lines 529-531). Our theorem (Theorem 5) shows a separation result. It essentially stated that VCG-based mechanisms, which is a wide class of known mechanisms, when equipped with output predictions and when requiring constant consistency, are restricted to quadratic robustness which is more limited than mechanisms with input predictions.
>
> **Our response to Question 2:** Those lemmas concern tight results for the proposed mechanisms. However, we also provide more general lower bounds for classes of mechanisms (that satisfy some natural properties), e.g. for all VCG-based in scheduling problem (Theorem 5) and for general mechanisms for the house allocation problem (Theorem 8). We remark that providing negative results for all mechanisms, even in the standard mechanism design setting (without any sort of prediction) may be notoriously hard, and only few such results exist (see e.g. [16] for scheduling, and [19, 36] for combinatorial auctions).
>
> **Our response to Question 3**: Yes, there exist such instances. The value of the two expressions in the inequality above depend on the value of $\hat{\rho}$, $\eta$ as a function of the output advice $\hat{a}$. For example, taking into account Lemma 5, it is possible that $\hat{\rho} = 1$, $\eta = 0$ and $\lambda$ is close to 1. The comparison of the two upper bounds is inconclusive, and this is the reason that we conduceted experiments (section B.5) in order to contribute to the better understanding of the relation between the two errors.
>
> **Our response to Question 4**: Actually, the reason is that Maximal in Range (MIR) compute the *optimal* allocation w.r.t social welfare among a subset of allocations (the range). One can enhance this subset of allocations by plugging-in the predicted output $a^*$. Then by definition of MIR, the MIR mechanism will output $a^*$ if this happens to be the best among the subset of allocations selected. Notice that this is a *deterministic* not a randomized mechanism. Studying randomized mechanisms with predictions is an interesting future direction.  However, any randomized mechanisms that outputs the prediction with constant probability cannot have the same robustness guarantee (in expectation) with the MIR mechanism we use.
>
>
> **Our response to Question 5**: Yes, this is exactly what we do in the facility location problem. Take the egalitarian cost for instance. The bound provided in [2] for the Minimum Bounding Box mechanism is tight. However, we manage to provide an even more refined upper bound in Theorem 1. This bound is tight (see Lemma 1), in the sense that there is no upper bound $\hat{\rho}-\epsilon$ *strictly* lower than the one we provide. As mentioned in Remark 1, this upper bound is tight whenever the output advice is inside the minimum bounding box.
>
> Concerning the minor comments, again thank you for catching this typo, and we will consider further defining MIR mechanisms, even though they are generally described in lines 972-974.

---

> > ### Comment · Reviewer_i6E2 · 2024-08-13
> >
> > Thank you for addressing all my further questions. I really enjoyed this conversation and would like to raise my score.
> >
> > Here are a few further responses that I did not expect the authors would reply as it is the end of the discussion period:
> > 1.  Response to Response to Question 2: So the tightness results indeed do not extend to general settings.
> > 2. Response to Response to Question 4: I look forward to seeing if there are more examples/general scenarios where the plug-and-play properties can apply.

---

### Author Rebuttal · Authors · 2024-08-06

We would like to thank the reviewers for their thorough reviews and thoughtful comments. We respond to each reviewer's questions with a separate reply to each of their reviews.

---

> ### Author Response · Authors · 2024-08-13
>
> We thank all the reviewers for showing their appreciation on our work and for their fruitful engagement during the discussion period which will certainly improve the presentation of our work .

---

### Decision · Program_Chairs · 2024-09-25

**Decision:**

Accept (spotlight)

**Comment:**

This paper proposes a new type of prediction for learning-augmented mechanism design: instead of predictions about the inputs to the mechanism, it considers predictions about the output. This new type of prediction is interesting, well-motivated, and novel. Strong results are achieved and this framework could also initiate some follow-up work.